# On the global reconstruction of ocean interior variables: a feasibility data-driven study with simulated surface and water column observations

Aina Garcia-Espriu<sup>1</sup>, Cristina González-Haro<sup>1</sup>, and Fernando Aguilar-Gómez<sup>2</sup>

**Correspondence:** Aina Garcia-Espriu (ainagarcia@icm.csic.es)

Abstract. This work uses data-driven approaches to study the feasibility of reconstructing ocean interior variables (temperature and salinity) from surface observations provided by satellites and interior observations provided by Argo floats. The feasibility of the approach is based on an Observing System Simulation Experiment (OSSE) in which we use the outputs from an ocean numerical model as the ground truth, and simulate a real observing system of the ocean, taking the surface of the model as a simulation of satellite observations, and vertical profiles in the same locations as the real Argo floats. We implemented different models based on Random Forest Regressors and Long-Short Term Memory networks, which were trained with the simulated observations and validated against the complete numerical model results. We obtain high spatial and temporal correlation using both technologies and an accurate description of the annual variability of the data, accompanied by small biases.

#### 1 Introduction

The ocean serves as Earth's primary climate regulator, functioning as a massive heat sink and carbon dioxide reservoir and distributing its thermal energy globally through currents (Webster, 1994). Ocean monitoring relies on two complementary approaches: satellite-based remote sensing, which provides extensive surface-level data, and in situ observations through programs like the Argo float network (Argo, 2009), which offers deep-water measurements up to 2000 m. Satellite missions such as SMOS (Kerr et al., 2010), Aquarius (Lagerloef et al., 2008), and SMAP (Entekhabi et al., 2010) provide global sea surface salinity (SSS) measurements, while AVHRR (Casey et al., 2010), MODIS (Kilpatrick et al., 2015), and Sentinel-3 (Donlon et al., 2012) deliver sea surface temperature (SST) observations. These measurements are processed into various operational products, including gap-filled SST products such as OSTIA (Good et al., 2020), MUR (Chin et al., 2017), and ESA CCI (Merchant et al., 2019), as well as SSS products, including BEC (Olmedo et al., 2021), SMAP Level 3 (Meissner et al., 2018), and ESA CCI (Boutin et al., 2021).

The accurate estimation of ocean state variables such as SST, SSS, and upper-ocean dynamics relies on a combination of in situ measurements, satellite remote sensing, and physical modeling. Physical and data-assimilative ocean models—ranging from general circulation models (GCMs) to regional high-resolution systems—enable the estimation of the full ocean column by solving the governing dynamical equations (e.g., Navier–Stokes, heat, and salt conservation), but are sensitive to initializa-

<sup>&</sup>lt;sup>1</sup>Institute of Marine Sciences (ICM), CSIC, Passeig Marítim de la Barceloneta, 37-49, 08003 Barcelona, Spain

<sup>&</sup>lt;sup>2</sup>Instituto de Física de Cantabria (IFCA), CSIC, Av. de los Castros s/n. Ed. Juan Jordá, 39005 Santander, Spain

tion and parameterization errors (Sloyan et al., 2019). Accurately resolving the first few meters of the ocean requires careful treatment of near-surface processes and vertical mixing, which are poorly constrained without surface-to-depth observations. In addition, resolving the full water column at the spatial and temporal resolution needed to capture near-surface dynamics entails considerable computational demands, particularly for data-assimilative systems. This challenge is faced in the Digital Twin of the Ocean (DTO) concept, where the application of Artificial Intelligence (AI) models to synthesize surface and deep-water measurements can achieve a 4D (3D ocean reconstruction + time variability) reconstruction of ocean dynamics. The integration of different data sources along with advanced modeling techniques presents a promising way to improve our understanding and prediction of oceanic changes in the context of global climate dynamics.

The integration of AI and data-driven methodologies has transformed physical oceanography, which was traditionally dominated by theoretical models and limited observational data. In recent years, oceanography has experienced a surge of interest in data-driven approaches, fueled by advances in computational resources, the availability of larger oceanographic datasets (e.g., from Argo, satellite missions, and reanalyses), and the growing maturity of deep learning frameworks. Notably, one of the earliest applications of neural networks in this field dates back to 2009, when Ballabrera-Poy et al. (2009) demonstrated the potential of simple neural architectures to improve the reconstruction of subsurface salinity profiles in the eastern North Atlantic. More recent research has demonstrated significant progress in reconstructing 2D and 3D ocean variables. Subsurface salinity has been reconstructed at  $0.25^{\circ} \times 0.25^{\circ}$  resolution in Tian et al. (2022) by enhancing coarser resolution products ( $1^{\circ} \times 1^{\circ}$  at a monthly scale). Subsurface temperature and salinity anomalies were reconstructed in Su et al. (2015, 2018, 2021) using monthly gridded Argo data and AI models such as Support Vector Machines and Long-Short Term Memory (LSTM) Neural Networks. They used monthly gridded Argo data at  $1^{\circ} \times 1^{\circ}$  resolution to perform their reconstruction. Buongiorno Nardelli (2020) proposed a deep learning network to reconstruct the ocean water column using combined satellite and in-situ measurements. Their innovative approach consists of using the potential of the LSTM not to predict a time series, but to predict a depth series. As in the previous studies, a gridded in situ dataset with monthly temporal resolution was used.

It has been demonstrated by the state of the art that different AI models can derive oceanic data at a large temporal scale and using gridded (and interpolated) in situ (*i.e*: 1-month Argo gridded products). Here, we want to assess if it is feasible to reconstruct the ocean interior variables at the most common native resolution of the microwave remote sensing products  $(0.25^{\circ} \times 0.25^{\circ})$  while maintaining a daily temporal resolution, which is suitable for studying temporal mesoscale ocean dynamics, combined with non-gridded in-situ data to avoid interpolations and to respect the spatial variability of the measurements.

The paper is structured as follows: in Sect. 2, we give an overview of the current sampling of the ocean from the perspective of the available products from remote sensing and in situ observations, in Sect. 3 we introduce the datasets used for the feasibility study. Then, in Sect. 4 we introduce the methodology, including an analysis of the limitations of the dataset, the selection of variables and model architectures, the generation of the simulated observation system, and the implementation and training of the models. The results are shown in Sect. 5 where we analyze the explainability, accuracy, and errors of the models' predictions with different tools and perspectives. In Sect. 6 we discuss the obtained results, and in Sect. 7 we summarize the conclusions and propose different paths to explore in this field.

#### 2 Current Sampling of the Ocean

#### 2.1 Surface Remote Sensing products

We collected key information of the most widely used sea surface level 3 (L3) and gap-filled level 4 (L4) products for some potential variables such as SSS, SST, SSH, and currents. The SSS is summarized in Table 1, dominated by SSS products from the SMOS and SMAP missions, but also from Aquarius (JPL, 2018). The earliest records are from the SMOS mission, starting in 2010/2011. It has two L3 products (ESA CCI v5.5 (Boutin et al., 2021) and BEC Global v2 (Olmedo et al., 2020a)) and one L4 (BEC Global v2 (Olmedo et al., 2020a)), but none of them are served operationally. Two L3 products are derived from SMAP (JPL v5.0 (NASA/JPL, 2020) and RSS v6.0 (Remote Sensing Systems, 2024)), starting from 2015 and offering an operational alternative. The uncertainty of all SSS products ranges between 0.2 and 0.5 pss, where 0.2 pss is the most common uncertainty value. The temporal resolution is approximately 1 week, and the most common spatial resolution is 0.25°.

The SST products are summarized in Table 2 (Good et al., 2020; Wentz et al., 2021; CMC, 2012; E.U. Copernicus Marine Service Information (CMEMS), 2024; NASA/JPL, 2015). All the products shown are derived from multi-sensor (microwave and infrared) analyses and are provided as daily Level 4 datasets, with the exception of *SST AMSR2 REMSS*, which is a Level 3 product. The spatial resolution varies among products, ranging between 0.01°, when combining infrared and microwave observations and 0.25° for the L3 microwave product. In any case, it is always equal to or finer than the ones from the SSS products, only available from microwave observations. For this reason, our target spatial resolution is 0.25°. The most common uncertainty value is 0.5°C. Most of the products overlap with the SSS time series, as the latest of the SST datasets begins in 2002 (SSS earliest product is from 2010). The only exception is the operational SMAP products, which do not overlap with ESA CCI and CMC SST datasets, which extend only until 2022 and 2017, respectively.

The SSH and surface current products are summarized in Table 3 (CLS (France), 2024; Dohan, 2021). All datasets are derived from multi-mission satellite altimetry and are provided as Level 4 gridded products, with daily or sub-daily temporal resolution. Spatial resolution varies between 0.25° and 0.1°, generally coarser than SST and SSS datasets. Typical uncertainty in SSH is around 2–3 cm, while current estimates have uncertainties of 5–10 cm/s depending on the region and method. Most products cover the full altimetry era beginning in 1992, thus overlapping entirely with the SSS time series. However, near-real-time products may lag in reprocessing quality and are not always consistent with the climate-oriented datasets such as DUACS or CMEMS reprocessed fields.

#### 2.2 In situ vertical profiles

The Argo program provides 300-400 daily vertical profiles worldwide. These profiles are not homogeneously distributed in space, as shown in Fig. 1. Each float has a 10-day cycle length during which it takes ocean measurements. The profiler rises slowly in the last 6 hours, taking measurements at different depths (2 meters being the most common sampling interval). Argo profiles report measurement uncertainties of approximately 0.002°C for temperature and 0.01 for salinity (Wong et al., 2023). The associated positioning error ranges between 0.25 km and 1.5 km in latitude and longitude, which is considered negligible for the purposes of this study. The coverage of the Argo floats is analyzed in Fig. 1. We observe how the total number of floats

**Table 1.** Gap-free Level 3 and Level 4 SSS Products Overview.

| Product name - Producer (References)                           |                           |                            |             |               |       |  |
|----------------------------------------------------------------|---------------------------|----------------------------|-------------|---------------|-------|--|
| Satellite(s)                                                   | <b>Spatial Resolution</b> | <b>Temporal Resolution</b> | Uncertainty | Time Period   | Level |  |
| ESA Level 3 SSS CCI v5.5 – ESA (Boutin et al., 2021)           |                           |                            |             |               |       |  |
| SMOS                                                           | $0.25^{\circ}$            | 1-week                     | 0.2-0.5 pss | 2010–2023     | L3    |  |
| BEC Global v2.0 L3 SSS – BEC (ICM/CSIC) (Olmedo et al., 2020a) |                           |                            |             |               |       |  |
| SMOS                                                           | $0.25^{\circ}$            | 9-day running mean         | 0.2-0.3 pss | 2011-May 2021 | L3    |  |
| BEC Global v2.0 L4 SSS – BEC (ICM/CSIC) (Olmedo et al., 2020b) |                           |                            |             |               |       |  |
| SMOS                                                           | $0.05^{\circ}$            | Daily                      | 0.2-0.3 pss | 2011-May 2021 | L4    |  |
| JPL SMAP Level 3 CAP SSS v5.0 – JPL (NASA/JPL, 2020)           |                           |                            |             |               |       |  |
| SMAP                                                           | $0.6^{\circ}$             | 8-day running mean         | 0.2 pss     | 2015-Present  | L3    |  |
| RSS SMAP Level 3 SSS v6.0 – RSS (Remote Sensing Systems, 2024) |                           |                            |             |               |       |  |
| SMAP                                                           | 0.25°                     | 8-day running mean         | 0.2 pss     | 2015-Present  | L3    |  |
| Aquarius CAP Level 3 – JPL (JPL, 2018)                         |                           |                            |             |               |       |  |
| Aquarius                                                       | 1°                        | 7-day running mean         | 0.2 pss     | 2011–2015     | L3    |  |

**Table 2.** Level 3 and Gap-free Level 4 SST Products Overview.

| Product name - Producer (References)                                               |                           |                            |             |              |       |
|------------------------------------------------------------------------------------|---------------------------|----------------------------|-------------|--------------|-------|
| Satellite(s)                                                                       | <b>Spatial Resolution</b> | <b>Temporal Resolution</b> | Uncertainty | Time Period  | Level |
| OSTIA Global SST – UKMO (Good et al., 2020)                                        |                           |                            |             |              |       |
| Multi-sensor                                                                       | $0.05^{\circ}$ (5 km)     | Daily                      | 0.3°C       | 1981-present | L4    |
| SST AMSR2 REMSS – RSS (Wentz et al., 2021)                                         |                           |                            |             |              |       |
| Multi-sensor                                                                       | $0.25^{\circ}$ (25 km)    | Daily                      | 0.5°C       | 2002-present | L3    |
| GHRSST Level 4 CMC Global Foundation SST v2.0 - CMC (CMC, 2012)                    |                           |                            |             |              |       |
| Multi-sensor                                                                       | $0.2^{\circ}$             | Daily                      | 0.5°C       | 1991 to 2017 | L4    |
| SST CCI – UKMO, ESA CCI (E.U. Copernicus Marine Service Information (CMEMS), 2024) |                           |                            |             |              |       |
| Multi-sensor                                                                       | $0.05^{\circ}$            | Daily                      | 0.5°C       | 1981 to 2022 | L4    |
| GHRSST Level 4 MUR Global Foundation SST v4.1– JPL (NASA/JPL, 2015)                |                           |                            |             |              |       |
| Multi-sensor                                                                       | $0.01^{\circ}$            | Daily                      | 0.5°C       | 2002-present | L4    |

Table 3. Sea Surface Height (SSH) and currents Products Overview

| Product name - Producer (References)                                                          |                        |                     |             |               |       |  |
|-----------------------------------------------------------------------------------------------|------------------------|---------------------|-------------|---------------|-------|--|
| Satellite(s)                                                                                  | Resolution             | Temporal Resolution | Uncertainty | Time Period   | Level |  |
| Global Ocean Gridded L 4 Sea Surface Heights And Derived Variables - CLS (CLS (France), 2024) |                        |                     |             |               |       |  |
| Multi-sensor                                                                                  | $0.25^{\circ}$ (25 km) | Daily               | 3-4 cm      | 1993-Nov 2024 | L4    |  |
| Ocean Surface Current Analyses Real-time (OSCAR) Surface Currents - PO.DAAC (Dohan, 2021)     |                        |                     |             |               |       |  |
| Multi-sensor                                                                                  | 0.25°                  | Daily/weekly        | 2-3 cm      | 2020-present  | L4    |  |

Figure 1. Distribution of the Argo profiles with at least 1000 m depth and good quality data from 2010 to 2022. The top row shows the spatial distribution at  $0.25^{\circ} \times 0.25^{\circ}$  and  $1^{\circ} \times 1^{\circ}$  regular grid. The bottom row shows the temporal distribution in a daily resolution (gray) and a 60-day mean sliding window (dark gray).

Figure 2. Argo floats coverage at  $1^{\circ} \times 1^{\circ}$  (top-left) and  $5^{\circ} \times 5^{\circ}$  resolution (top-right). Intra-pixel standard deviation of the salinity (bottom-left) and temperature (bottom-right) measurements as seen by the reanalysis model at  $5^{\circ} \times 5^{\circ}$  resolution.

in a pixel of the most common satellite resolution  $(0.25^{\circ} \times 0.25^{\circ})$  for the 13-year time series, provides, on average, between 5 and 10 measurements (top-left plot of Fig. 1). At a spatial resolution of  $1^{\circ} \times 1^{\circ}$ , the complete time series yields an average of 40 to 80 measurements per pixel (top-right panel of Fig. 1), which represents a relatively low sampling density in both cases. When observing the temporal evolution in the number of floats in the bottom plot in Fig. 1, we see that there are consistently more than 400 unique floats per day between 2012 and 2020. For the remaining period, there is a slightly smaller number of floats, which are, on average, 300 per day. To see how it applies to our desired reconstruction temporal resolution (daily, 10-day mean sliding window), we show in Fig. 2 the number of Argo floats for a 10-day window (which corresponds to the profiler cycle length and is a similar value of the aggregation window of multiple satellite sources) at  $1^{\circ} \times 1^{\circ}$  and  $5^{\circ} \times 5^{\circ}$  resolution (top columns). Then, for the coarser resolution, we computed the intra-pixel standard deviation of the salinity and the temperature as seen by the reanalysis model. We observe that, to obtain almost complete coverage of the globe, we would need to resort to a resolution of  $5^{\circ} \times 5^{\circ}$  as seen in the top-right panel in Fig. 2. The intra-pixel variability of each  $5^{\circ} \times 5^{\circ}$  box is shown in the bottom panels in Fig. 2 and can reach values of more than  $1.5^{\circ}$ C in temperature and  $0.5^{\circ}$  (g/kg) in salinity. This variability would worsen the reconstructions in areas where the intra-pixel variability is very high (which are the regions of more activity and thus, zones of interest).



#### 105 3 Datasets





The data-driven approach presented in this study relies on two complementary datasets. The first one is the in-situ measurements from the international Argo float network, which provides vertical profiles of the physical properties of the ocean (temperature and salinity). The second dataset is the Copernicus Marine Service Global Ocean Ensemble Reanalysis product, produced with a numerical ocean model constrained with data assimilation of satellite and in situ observations. It provides a complete picture of ocean state variables. The specific products that are used in the study are:

- Argo floats. We use Argo profilers (Argo, 2025) to determine the positions that our simulated observing system has to sample. Argo data are collected and made freely available by the International Argo Program and the national programs that contribute to it (http://www.argo.ucsd.edu). The Argo Program is part of the Global Ocean Observing System. We use all available profiles from 2010 to 2022, but only consider those that reach a minimum depth of 1000 meters and have good quality measurements according to their quality control standard.
- Copernicus Marine Service Global Ocean Ensemble Reanalysis. This study has been conducted using E.U. Copernicus Marine Service Information (https://doi.org/10.48670/moi-00024). The Copernicus Marine Service Global Ocean Ensemble Physics Reanalysis product (Mercator Océan International, 2025) is given at 0.25° × 0.25° resolution and contains daily temperature, salinity, currents, and ice variables for 75 vertical levels. We use this reanalysis to simulate sampling properties of both in situ observations and satellite surface data.

#### 4 Methodology

This work uses a reanalysis of daily 3D gridded data to simulate the current ocean observation system and assess the feasibility of a 4D reconstruction of the ocean variables. Using a reanalysis model instead of in-situ data enables us to access the locations that are not sampled by the in-situ measurements. The sampled locations are used in the training datasets of our models (both in the train and test splits). In contrast, the unsampled locations can be used to validate how the model extrapolates to regions not seen by the profilers.

We discussed in Subsect. 2.2 how the small number of in situ measurements could affect our reconstructions. This poses a problem when trying to reconstruct the daily state of the ocean. So, to be able to work with daily data, we opt to use a sparse data approach instead of gridded datasets, working with individual profiles. With this methodology, the input of the models is a structure that contains the surface variables along with the acquisition condition identifier (*i.e.* latitude, longitude, time of the year) as predictors and the salinity and temperature vertical profiles as predicted variables. The following sections describe which are the potential variables and how the final OSSE dataset is generated. We also discuss which kind of AI models we apply for this experiment and how they were implemented.

## 4.1 Potential predictor variables




Satellite observations offer measurements of multiple variables that can be interconnected and describe different processes of interest. However, not only do the measured values offer important information, but also their acquisition conditions, such as the acquisition time or the geolocation. As this information also helps in the description and modeling of the oceanic processes, considering them in our models can help in the understanding of the relationships between different water masses or seasonal patterns. In this work, we consider the observation acquisition conditions to be the latitude, longitude, depth, and time of the measurements. For example, sea surface temperature shows a latitudinal variation pattern, with the equatorial region being the warmest. It also presents daily and seasonal cycles, both of which present latitudinal variations. The longitude coordinate does not directly impact the measured variables, but, along with the latitude, can help the model understand the Earth's topology and determine the position of different water masses. The depth of the observations also affects their measured values, as in the deeper layers, the ocean processes occur at a much larger time scale. In contrast, in the upper layers, the time scales of the oceanographic processes rapidly decrease, and the variation is greater. Finally, the observation time affects the measurements at different scales, as both salinity and temperature present a seasonal cycle, so knowing at which time of the year the measurement was taken can give information on those patterns.

The surface measurements include SSS, SST, sea surface height (SSH), mixed-layer depth (MLD), and information about the currents, which are represented as the zonal and horizontal components of the velocity vectors (UO and VO, respectively). The combination of SSS and SST defines the sea surface density. These relations contribute to the stratification and vertical mixing of the ocean, which in turn affects MLD stability. Increased heating at the surface can make the MLD shallower, while increased freshwater input can make it deeper. SSH provides critical information about the ocean's dynamic topography and large-scale circulation patterns. It reflects the integrated effect of temperature, salinity, and pressure over the water column and is closely linked to the geostrophic flow. Similarly, surface currents (UO and VO) are essential indicators of advective processes and mesoscale dynamics that transport heat and salt laterally across the ocean. These dynamic variables provide context to the surface forcing and enhance the ability of statistical or machine learning models to infer subsurface structures from sparse observations.

The variables that we use in the training of the models are a combination of the aforementioned ones. However, one key aspect when selecting the model predictors is knowing the limitations of the data. The number of input variables can positively and negatively affect the model's outputs. On the one hand, if too few variables are used, we may not be able to describe the oceanographic processes of interest. On the other hand, if too many variables are used, the model will not have enough data to describe all the possible combinations, and its quality will rapidly degrade. Thus, the specific selection of the variables used on each model will be further discussed in Subsect. 4.4.

## 4.2 Simulated profiles construction

To simulate the current observation system, we need to determine the positions in which an in-situ observation was taken and retrieve the equivalent data from the reanalysis oceanographic global circulation model dataset (daily 3D gridded product in

latitude, longitude, and depth), as described in Fig. 3. First, we determine in which positions we had in-situ measurements and need to be part of the simulated dataset. Positions marked with an "Argo float" in Fig. 3 represent positions sampled by Argo floats that pass the following filtering criterion. We select Argo profiles that measure at least up to 1000 m depth to guarantee homogeneity in the input dataset and to ensure that the profiles have measurements in the complete interest range of the water column. Furthermore, it removes the points close to the coast that typically present different dynamics than the open ocean. To guarantee the quality of the observations, we only use profiles that offer the variables given in adjusted mode that have the best possible quality control (see "1: Good" in (Argo Data Management Team, 2024), pg. 105). Curated and calibrated profiles give a more faithful representation of the measurements. Finally, we only use profiles contained in the 60 °S - 60 °N latitude range as the polar region's dynamics greatly differ from the open ocean ones.

Once we have the list of sampled positions, we take the complete water column from the reanalysis up to 1000 m. The ocean presents most changes in the first meters, and becomes more stable in deeper waters. We use the same vertical sampling as the reanalysis model, which becomes coarser at greater depths. Then, for each date, we join the 10-day sliding window simulated profiles with the simulated surface data of the central date (SST, SSS, currents, SSH, MLD) by adding to each profile the values of the first layer of the reanalysis variables at that same location. We used the reanalysis at  $0.25^{\circ} \times 0.25^{\circ}$  resolution to match the coarser satellite data grid resolution, so in this step, no interpolation or merging is required. Then, in order to take into account the uncertainty of the current observing systems, we add a Gaussian distribution with zero mean and standard deviation equal to the estimated uncertainty to each measurement (see Table 4). It must be noted that in real-world measurements, the uncertainty values can present more consistent geophysical patterns, having, for example, higher values near the coast or in high latitudes. Thus, modeling the uncertainties as Gaussian noise serves as a first approach to quantifying the uncertainty propagation in our analysis, though it may not capture the full complexity of real-world error structures. However, for real data applications, further work would be needed to consider uncertainties that might have spatially correlated structures, temporal dependencies, or non-Gaussian distributions, and that are usually provided by the data producers (Merchant et al., 2019; Olmedo et al., 2021). We store the datasets in daily files, whose structure and uncertainty values are shown in Table 4.

Finally, we separate our datasets into a train/test split, which will be common for all the trained models. This separation is made using an 80/20 ratio, where 80% of data will be used for training and 20% for validation as usual in machine learning models. We generate one dataset (or datafile) for each day. As the objective of our study is to analyze the feasibility of the reconstruction using current sampling of the ocean (and not predicting future trends and events), the separation is done by randomly separating the dates, but ensuring that each month is represented equally in both datasets. This avoids adding imbalances due to seasonal cycles that must be accounted for.

# 4.3 Models selection






Two model architectures were chosen for this study: the Random Forest Regressor (RFR) (Breiman, 2001) due to its algorithmic simplicity and training cost and the Long-Short Term Memory (LSTM) (Hochreiter and Schmidhuber, 1997) due to its already demonstrated utility in oceanographic applications.

**Table 4.** Simulated OSSE dataset. The dimensions of the dataset are "id", which corresponds to an index term to identify the profiler, and "n\_depth," which corresponds to the depth level index. The vertical profiles are constructed using the 10-day sliding window data, and the surface data is from the central date of the window.

| Variable name   | Туре    | Dimensions             | Uncertainty |  |  |
|-----------------|---------|------------------------|-------------|--|--|
| Predictors      |         |                        |             |  |  |
| Latitude Index  | Integer | <id>&gt;</id>          | -           |  |  |
| Longitude Index | Integer | <id>&gt;</id>          | -           |  |  |
| Day of the year | Integer | <id>&gt;</id>          | -           |  |  |
| Depth           | Decimal | <n_depth></n_depth>    | -           |  |  |
| SSS             | Decimal | <id></id>              | 0.2 pss     |  |  |
| SST             | Decimal | <id></id>              | 0.5 °C      |  |  |
| SSH             | Decimal | <id></id>              | 3 cm        |  |  |
| MLD             | Decimal | <id></id>              | 1 m         |  |  |
| UO              | Decimal | <id></id>              | 0.05 m/s    |  |  |
| VO              | Decimal | <id></id>              | 0.05 m/s    |  |  |
| Predicted       |         |                        |             |  |  |
| ASAL            | Decimal | <id, n_depth=""></id,> | 0.01 pss    |  |  |
| СТЕМР           | Decimal | <id, n_depth=""></id,> | 0.002 °C    |  |  |

On one hand, the RFR can model non-linear relations between the variables, which is a key aspect when studying oceanographic processes. Furthermore, it does not need large datasets to produce good predictions. It works by constructing multiple decision trees during training and outputting the mean prediction of all trees for a more robust result, as illustrated in Fig. 4. Each tree is built using a random subset of the training data and considers a random subset of features at each split point. The randomness helps prevent overfitting and the averaging of the predictions gives stability and accuracy in the results.



On the other hand, LSTM architectures have proved to produce promising results in the field (Buongiorno Nardelli, 2020; Su et al., 2021). This type of architecture can handle long-term dependencies while maintaining stable gradients and mitigating the vanishing gradient problem. LSTMs can remember important information and forget irrelevant details through three main gates: the forget gate decides what information to discard, the input gate determines what new information to store, and the output gate controls what parts of the cell state should be output. It is well-suited for tasks that generate data sequences, as in this case, where a vertical profile is generated for each surface data point. By comparing these two models, we can not only determine which one has better performance but also discuss whether we have enough data to use deep learning mechanisms and if the improvement regarding simpler models is worth the cost.

# **Synthetic Datasets Generation**

**Figure 3.** Observing system datasets generation from Copernicus Marine Service reanalysis data using Argo floats and surface satellite measurements sampling. For each day, the 10-day windowed simulated profiles are collocated with the central date data, generating a daily array of synthetic profiles. The different colors indicate different days in the 10-day window of a specific central date.

#### 4.4 Implementation and Training



We implemented different models using RFR and LSTM architectures, varying the input variables and the configurable parameters such as the number of trees, the number of layers, the number of units in each layer, etc. However, in this work, we only present the two RFR and two LSTM configurations that are interesting for the discussion. In the case of the RFR, we trained one model (RFRv1) with only the SSS and the SST as surface variables to see to what extent the interconnection of these variables could predict their vertical profiles. Then, we trained another model (RFRv2) with the complete set of surface variables to compare it with the predictions made by the LSTM models.

In the case of the LSTM, we trained different models varying their architecture configuration and hyperparameters, shown in Fig. 5, such as the activation function, the number of layers, the number of units, the learning rate, etc. For the discussion, we selected the same configuration used in Buongiorno Nardelli (2020) to check whether it could be extrapolated to our problem (LSTMv1). We also selected the best-performing salinity and temperature predictor (LSTMv2). The exact set of predictors and tuning parameters of each model are the ones as follows:

**Figure 4.** Random Forest Regressor structure. The final prediction is the average of all individual tree predictions, reducing overfitting and variance compared to using a single decision tree.

- **RFRv1:** Salinity and Temperature as surface variables. Maximum of 100 decision trees, a minimum of 10 measurements per leaf, and a maximum tree depth of 20.
  - **RFRv2:** Salinity, Temperature, Currents, MLD, SSH, and latitude as surface variables. Maximum of 100 decision trees, a minimum of 10 measurements per leaf, and a maximum tree depth of 20.
  - LSTMv1: Salinity, Temperature, Currents, MLD, SSH, day of the year, depth, longitude, and latitude as surface variables. Two LSTM layers with 35 units, a dropout value of 0.2, a learning rate of 1e-5, early stopping, and the hyperbolic tangent (tanh) as the activation function. Trained with at least 100 epochs using Early Stopping functions to avoid overfitting.



LSTMv2: Salinity, Temperature, Currents, MLD, SSH, day of the year, depth, longitude, and latitude as surface variables. Three LSTM layers with 1024 units, a dropout value of 0.2, a learning rate of 1e-5, and the Softsign as the activation function.

The models in this work are designed to conduct a reconstruction task. We reflect this in the separation of the training and test datasets. If the model had to predict future events, we would need to divide some consecutive years for training and the rest

**Figure 5.** Model architecture using LSTM layers. The model takes as input a list of points to predict, where each point is a matrix of  $n\_depths \times n\_vars\_in$  positions. Then, it connects with  $n\_layers$  groups of LSTM + Dropout layers. Finally, a Time distributed Dense layer that produces the final predictions.

for testing. In this case, we aim to assess if the model can reconstruct what it has already seen on the surface, so the datasets are randomly divided on a daily frequency. The datasets are balanced monthly and yearly to avoid possible biases due to the division imbalance. Furthermore, we use the same train/test splits for all the trained models to ensure that the ingested data is the same. Both models are implemented in Python (Van Rossum and Drake, 2009) using standard libraries such as Tensorflow (Abadi et al., 2015) and Scikit-learn (Pedregosa et al., 2011). The datasets are preprocessed from the original netCDF (Rew et al., 1997) files to Feather format, which is a column-oriented binary disk-based format based on Apache Arrow and supported by Python. This optimizes the data ingestion, which can be one of the major bottlenecks in the training process.

# 245 5 Results



We ran the four models with the complete test dataset (which is common for all models), both using training data with Gaussian noise (matching uncertainty values) and without it. Table 5 shows each model's accuracy (R<sup>2</sup>) and error metrics (MSE, MAE), both in temperature and salinity predictions. When not contemplating uncertainties estimates, we obtain accuracies that range between [0.76 - 0.96], making the RFRv2 and LSTMv2 better than their v1 counterparts. When the uncertainty error is accounted for, we observe a slight degradation in the performance of the models, with an accuracy ranging between [0.75 - 0.94]. The most sophisticated models still perform better than their simpler counterparts. In both cases, the salinity predictions are more accurate and less prone to errors than the temperature ones. From now on, all the results correspond to the models trained with uncertainty estimates unless noted otherwise.

| Model   | R <sup>2</sup> (SAL//TEMP) | MSE (SAL//TEMP)    | MAE (SAL//TEMP) |
|---------|----------------------------|--------------------|-----------------|
| RFRv1*  | 0.88 / 0.76                | 0.08 / 5.70        | 0.17 / 1.49     |
| RFRv1   | 0.86 / 0.75                | 0.11 / 5.83        | 0.20 / 1.58     |
| RFRv2*  | 0.95 / <b>0.85</b>         | 0.04 / 3.16        | 0.11 / 0.86     |
| RFRv2   | 0.94 / 0.84                | 0.11 / 3.43        | 0.14 / 1.01     |
| LSTMv1* | 0.87 / 0.82                | 0.08 / 4.03        | 0.20 / 1.17     |
| LSTMv1  | 0.92 / 0.79                | 0.09 / 4.57        | 0.22 / 1.28     |
| LSTMv2* | <b>0.96</b> / 0.84         | <b>0.04</b> / 3.76 | 0.13 / 1.01     |
| LSTMv2  | <b>0.94</b> / 0.79         | <b>0.06</b> / 4.58 | 0.17 / 1.16     |

**Table 5.** R<sup>2</sup>, mean squared error, and mean absolute error metrics for salinity and temperature for the different models. The best-performing model is highlighted in bold text for each metric. The asterisk indicates that the model does not account for uncertainties in the measurements.

Figs. 6 and 7 show the same metrics as in Table 5 but as a function of the depth of the measurement. Salinity models display a robust prediction between depths, as the slope of the functions/metrics with respect to depth is almost vertical. There, we can observe that RFRv2 provides a higher R<sup>2</sup> and smaller errors than the rest of the models, closely followed by the LSTMv2 approach. In the case of temperature, we can see that metrics decline at about 200 m depth, where the predictions start to fail and errors grow larger. In salinity, we can also observe the slope change to a lesser extent, but the scores remain stable in the lower depths. Reconstructing the temperature field proves more challenging and requires further work to achieve the quality of the salinity reconstruction. In Appendix A, we provide the complete validation of the temperature, but due to its lower quality when compared to the salinity reconstruction, we decided to focus on the validation of the salinity from now on.

#### 5.1 Model explainability



Machine learning-based models, as opposed to physical models, pose a problem when interpreting the results obtained, as they are sometimes treated as black boxes where we know the input information and the output produced. Still, we do not understand the processes involved in obtaining the results. Understanding how these outputs are made helps us understand the causes and improve the models by focusing on the essentials. Although Random Forest-based models are more transparent and interpretable than those based on LSTM networks, a method of interpretability comparable to both has been chosen to improve the analysis.

SHAP (SHapley Additive exPlanations) (Lundberg and Lee, 2017) is a framework for interpreting machine learning model predictions based on cooperative game theory principles. It is based on the Shapley value concept from game theory to allocate contributions of individual features to the output of a model fairly and consistently. They quantify how much each feature contributes to moving the model prediction from a baseline (typically the mean prediction) to the actual output for a specific data point.

The resilience to noise varies among model architectures. This can result in some variables not being used if the model is not able to extract information from them. In order to reflect how the uncertainties affect the information the models can extract for

**Figure 6.** Predicted salinity R<sup>2</sup>, mean squared error, and mean absolute error (from left to right) per depth for the different models: RFRv1 (blue), RFRv2 (orange), LSTMv1 (green), LSTMv2 (red).

Figure 7. Predicted temperature  $R^2$ , mean squared error, and mean absolute error (from left to right) per depth for the different models: RFRv1 (blue), RFRv2 (orange), LSTMv1 (green), LSTMv2 (red).

each variable, we computed SHAP values with and without uncertainties. In order to see how the contribution of each variable changes in the vertical profile, SHAP values are computed for each depth of the predictions. Fig. 8 shows the percentage of each variable contribution for the RFRv2 and LSTMv2 models, when uncertainties were not accounted for in the training data. We only show these two models because they are the most complex and should be the "ideal case". For the RFR (Fig. 9), we observe that the SSS is the dominant predictor, which provides a climatological reference. Then, variables like SST, latitude, and SSH contribute by capturing temporal and regional variations in the signal. The LSTM model (Fig. 10), however, shows a different learning strategy as it does not rely on SSS. Instead, it derives from other variables such as longitude and temperature. The latitude variable is also mostly accounted for in the inherent temperature's latitudinal structure. This approach captures both the climatological baseline and dynamic components without depending directly on SSS measurements.

We compare how the variable selection and importance change when uncertainty is included in the training data. SHAP results for the RFR models (with uncertainties) are shown in Fig. 9. Both RFRv1 and RFRv2 use the SSS as the primary predictor variable, followed by the SST. We do not observe remarkable differences when compared to the results obtained without accounting for the uncertainties (Fig. 8). This is because the RFR is an ensemble model that has high tolerance to noise. The SHAP values for the salinity LSTM model are shown in Fig. 10. We observe a similar pattern in the SSS, SST predictors. Both variables have more impact in the first 100 m. After that, their weight decreases (substituted by SSH) and becomes stable. Unlike in Fig. 8, the LSTM models are not able to extract information from the MLD and the currents. The latitude variable is also mostly erased from the prediction. This can be due to the spatial roughness induced by the randomness of the uncertainties, as the model faces problems when searching for a smooth relationship with spatial patterns.

This analysis shows the fundamental differences between traditional machine learning and deep learning approaches in ocean reconstruction applications. While RFR excels at direct variable relationships, LSTM infers complex spatiotemporal patterns from indirect indicators. This capability becomes particularly valuable in regions with sparse SSS measurements or when reconstructing historical salinity patterns, where direct observations may be limited. However, LSTM is less resilient to noise, so errors in the input datasets should be handled with care.

#### 5.2 Validation with the test split dataset





To gain insights into the vertical reconstruction, we show in Fig. 12 the reconstructed vertical profiles of the points shown in Fig. 11 for the 29th October 2022, compared to the ground truth. We can observe a better fit in the case of the RFRv2 and LSTMv2 as they present a good correspondence with the ground truth curve. The models present some difficulties when predicting sharp transitions, as seen in the third and fourth points, as they tend to smooth the vertical profile. Sharp transitions, however, can be due to the quality of the numerical model or a well-mixed layer that produces abrupt changes. Further studies should be conducted in these difficult regions when using real in-situ data, and this metric should be revisited.

We have used the test split dataset aggregated as daily  $5^{\circ} \times 5^{\circ}$  maps (to have complete coverage) to validate the reconstruction's spatial coherence. We aggregated the predicted profiles in each grid cell and subtracted the aggregation of the true values in the same cell. In Fig. 13 we can see an example for the 29th of October 2022. We can observe low biases, which are about  $\pm 0.2$  g/kg in the major part of the map for all models except LSTMv1, which displays a latitudinal pattern. In broad strokes, the

Figure 8. Feature importance percentage using SHAP values for RFv2 (left) and LSTMv2 (right) salinity models without uncertainty values.

**Figure 9.** Feature importance percentage using SHAP values for RFRv1 (left) and RFRv2 (right) salinity models. Both models are trained with data that includes uncertainties.

bias patterns are similar among the different model configurations and are in the range of observed biases in satellite products of the same resolution for this region as reported in Figure 6 of Olmedo et al. (2021).

Through this first validation, we saw that the RFRv1 misses key surface variables that can potentially improve the quality of the reconstruction. Variables such as SSH and latitude have a high contribution to the predictions as seen in the SHAP values of the RFRv2 (Fig. 9, right panel). We also saw that the architecture presented in Buongiorno Nardelli (2020) requires some fine-tuning to optimize its performance to the given challenge. In subsection 5.3, we conduct a more in-depth validation of the best-performing models (RFRv2 and LSTMv2).

**Figure 10.** Feature importance percentage using SHAP values for LSTMv1 (left) and LSTMv2 (right) salinity models. Both models are trained with data that includes uncertainties.

**Figure 11.** Mixed-Layer Depth as seen by the reanalysis on the 29th October 2022. Points used in the validation are shown in Figs. 12 and A4 are referenced here in white.

ASAL (g/kg) at [-40.445°, 30.86°] ASAL (g/kg) at [29.981°, -50.088°] ASAL (g/kg) at [-128.586°, -0.948°] ASAL (g/kg) at [102.329°, -35.791°]

**Figure 12.** Predicted salinity vertical profiles for four different points (see Fig. 11 for its location) and the different models: RFRv1 (blue), RFRv2 (orange), LSTMv1 (green), LSTMv2 (red). The date is 29th October 2022. The black line corresponds to the OGCM baseline (the target value).

Figure 13. Predicted salinity minus Ground Truth (50 m depth) at a  $5^{\circ} \times 5^{\circ}$  grid (from top to bottom and left to right): RFv1, RFv2, LSTMv1, LSTMv2. The date is 29th October 2022.

## 5.3 Validation with the reanalysis dataset







We chose the region with longitudes ranging between 80 °W and 40 °W and latitudes ranging between 25 °N and 44 °N as it is a highly dynamic area that comprises the Gulf Stream current (see Figure 2). Notice how the region's sampling is non-homogeneous and focuses on more dynamical regions, *i.e.*, where the Gulf Stream is located. The validation is conducted with the predictions of both RFRv2 and LSTMv2 for the 2008-2009 time period, as it does not overlap with the training dataset. Three different levels of depths are considered to give an insight into the column variability: 5, 50, and 500-meter depths.

First, in Fig. 15, we assess the temporal biases of the 2-year averaged regional maps by computing the difference between predicted values and the reanalysis (ground truth) for the selected depths during both years. We can observe that the biases are in the 0.1-0.2 pss range at most. The high biases observed in the region close to the coast in Fig. 15 are because the model predicted values outside of the training region (profilers did not reach the 1000 m specified in the data preparation, as seen in Fig. 14).

Then, in Fig. 16 we show the temporal variability seen by the numerical model (our ground truth) and we use it as a reference to assess the differences with our predictions. We performed a significance test using a Fisher F-test for the difference of the variances at a 95% confidence level. Areas with significant differences are indicated as contours. The models are losing a variability range of about 0.1 g/kg and display spatial patterns similar to the ones observed in Fig. 15. The LSTMv2 model has more regions where the variability differences are not significant, mostly in the first layer, but overall, the differences are statistically too large. However, the differences do not grow larger in highly dynamic areas, implying that the models can capture the dynamical fluctuations of the salinity through the time series (although with less intensity), and thus, it separates from the climatological value of the predicted variable.

The temporal mean squared error (MSE) helps to understand which water masses the model has more difficulty representing. A zone with a high MSE could indicate that the model has not learned how to describe its dynamics, which can be due to undersampling of that region or highly complex dynamics that the model was not able to learn. Figure 17 shows the temporal MSE for both models, which have small values up to 0.1 (g/kg). This low value means that both models have a good characterization of the dynamics. In the case of the RFRv2, the maps are more stable in terms of MSE both horizontally and vertically, but the values are low on both models. As also observed in Fig. 15, the high MSE values observed in the region close to the coast are because we are predicting values outside of the training region, and should be discarded.

In Fig. 18, we compute the temporal correlation coefficient, computed as the Pearson correlation between the proposed model and ground truth time series for each gridpoint. This metric provides information about the respective models' capability to describe the variables' temporal cycles properly. As shown before in Fig. 16, the models can capture the same spatial pattern of temporal variability seen by the reanalysis (ground truth). However, we further analyze the temporal correlation to quantify the extent to which both the reconstructed and the ground truth show the same temporal variability evolution. As can be seen, both models properly describe the temporal variability of the reconstructed salinity for the shallower layers (5 m and 50 m). For the deeper layer considered here (500 m), the reconstruction is degraded in the southern part of the region, as we separate

**Figure 14.** Number of Argo profiles at 1/100 region-size resolution from 2010 to 2022 in the region where the models have been compared to the reanalysis product (black rectangle).

from the more dynamic region of the Gulf Stream Current. We further analyzed the impact of MLD on the reconstruction to check to what extent a deeper mixed layer provided better reconstruction, but no conclusive results were found (not shown).






It is also important to remark on the artifact that we can observe in Fig. 17 in the RFRv2 at a 5-meter depth (also visible in other plots). Horizontal lines appear due to the inclusion of the latitude coordinate in the model. Although RFRv2 produces great statistics (the variability is well captured, the MSE is low, and the correlation is high), it also produces artifacts in the latitudinal direction due to the binarization of the decision trees. This should be further investigated if we want to use this model architecture in the future.

Finally, we compute the temporal series of the spatial correlation coefficient to understand if there is any temporal or seasonality variation in the performance of our models. For each studied depth, the time series is smoothed using a 30-day moving average. We can observe in Fig. 19 that both models present a good correlation at all studied depths, with an R<sup>2</sup> score higher than 0.85. In the case of the RFR, we do not observe any tendency in the data or seasonal biases, meaning that our model can capture both possible tendencies and seasonal variability of the reconstructed variable. In the case of the LSTM, we observe some peaks of lower correlation that match the months in which the MLD is shallower, for this region during July. This result is consistent with the reconstruction of interior ocean variables using dynamical approaches. Within the framework of Surface Quasi-Geostrophy (SQG), it is possible to infer subsurface density and velocity anomalies from surface buoyancy fields, under the assumption that these anomalies originate at the surface and decay exponentially with depth Held et al. (1995). The depth of the mixed layer (MLD) plays a critical role in modulating the penetration of surface-driven anomalies into the ocean interior. Studies by Isern-Fontanet et al. (2008) and Miracca-Lage et al. (2022) demonstrated that SQG-based reconstructions can effectively capture subsurface structures—such as eddies and fronts—using only surface observations, especially when the MLD is deep and the surface stratification is weak. Conversely, when the MLD shoals, the surface signal becomes less representative of interior dynamics, highlighting the importance of accounting for MLD variability in data-driven ocean diagnostics and reconstruction methods.

**Figure 15.** Biases at different depths (from left to right: 5 m, 50 m, and 500 m) between predicted salinity and ground truth for the two proposed models, RFRv2 (top row), LSTMv2 (bottom row).

**Figure 16.** Temporal variability assessment between the predictions and the ground truth (reanalysis) salinities at different depths (from left to right: 5 m, 50 m, and 500 m). The standard deviation of the reanalysis (taken as our ground truth) for the 2008-2009 period (first row). Differences between the standard deviation of the ground truth and each model, RFRv2 (second row), and LSTMv2 (third row). Solid lines delimit statistically significant areas at a 95% confidence level using an F-test for the difference of variances.

**Figure 17.** Mean Squared Error at different depths (from left to right: 5 m, 50 m, and 500 m) between predicted salinity and ground truth for the two proposed models, RFRv2 (top row), LSTMv2 (bottom row).

**Figure 18.** Temporal Correlation Coefficients at different depths (from left to right: 5 m, 50 m, and 500 m) between ground truth and predicted salinity for the two proposed models RFRv2 (top row), LSTMv2 (bottom row).

**Figure 19.** Salinity spatial correlation time series using a moving average of 30 days at different depths (5m black solid line, 50 m black dashed line, and 500 m black dotted line) for the two proposed models for RFRv2 (left) and LSTMv2 (right). The temporal evolution of the average mixed layer depth is superposed in blue.

#### 5.4 Spatio-Structural Validation

As a final test, we perform a singularity analysis (Turiel et al., 2008) of our reconstruction to assess the extent to which the different models keep the spatial coherence and the geophysical consistency of the ground truth. Singularity analysis has previously been applied to evaluate the geophysical consistency between different datasets (Hoareau et al., 2018; Olmedo et al., 2021). Singularity exponents characterize the rate of change in oceanographic variables—such as SST, SSS, or surface velocity—across space. These exponents are particularly useful for detecting and quantifying sharp gradients, including features like fronts, eddies, and currents. In this context, the strongest fronts are associated with the smallest singularity exponent values (white lines in Figure 20), and potentially represent the streamlines of the flow, providing information about ocean circulation (Turiel et al., 2009). Here, we illustrate in a qualitative way how the LSTMv2 gives a better feature reconstruction than the RFRv2 when comparing with the OGCM ground truth. For example, the front meander located at 37.5°N and 63°W present in the ground truth is well reconstructed for the LSTMv2, but the RFRv2 gives a closed eddy.

# 6 Discussion




We implemented two approaches to study the feasibility of the 4D ocean reconstruction using the actual sampling capabilities provided by satellite and in-situ profilers, each of them with their own challenges and insights. The RFR exhibited artifacts due to the inclusion of latitude among input variables, suggesting that alternative techniques, such as incorporating neighboring region measurements instead of geographical coordinates, might be more effective for spatial contextualization. However, the

**Figure 20.** Singularity analysis of the salinity reconstruction at 500 m depth. Maps are from February 1st, 2008. Salinity (first column) and its associated singularity exponents (second column) for: the reanalysis without uncertainties (first row), RFRv2 (second row), and LSTMv2 (third row).

representativeness of the data might not be sufficient with the current ocean sampling, as the high number of predictors relative to the limited spatial and temporal coverage of marine observations could lead to overfitting in the random forest model, potentially reducing its ability to generalize to undersampled regions or periods. In the LSTM approach, we explored various configurations by adjusting the number of layers, activation functions, and units per layer. Both architectures demonstrated particular strength in salinity reconstruction, achieving high accuracy in the first and intermediate depths. However, correlation decreased in deeper layers with minimal ocean variations (specially for the case of temperature profiles, see Figure 7), where climatological values might suffice. Temperature reconstruction showed superior results with the RFR compared to LSTM. Nevertheless, the artifacts produced by input variables limit their application in 4D reconstruction. The RFR might still be valuable in studies with less critical horizontal dimensions. The dependence of the reconstruction metrics with the depth has also been reported previously by other authors that attempted to retrieve subsurface temperature anomalies using satellite-based data and gridded Argo in situ observations (Su et al., 2018). They obtain a lower  $R^2$  metric (< 0.72) than in our study but in their case this value is more homogeneous with depth until 500-m where it decreases to 0.5 for October (their worst case), similar values than in our case (see Figure 7). However, some considerations need to be taken into account when comparing both works. The studied region in Su et al. (2018) is the Indian Ocean, and in our case, the training is done at a global scale. Another difference is the spatial and temporal resolution of the training dataset; they used a monthly  $1^{\circ}$  gridded Argo dataset,

whereas we use individual profiles at the Argo buoy location. It is also important to highlight that in our approach, the individual profiles are only used for training the models, and the reconstruction is performed using just surface observations.

The validation with the test split dataset demonstrated that an increased number of surface variables improved the reconstruction of the water column, as the vertical profiles adjusted more faithfully to the ones of our ground truth and the spatial biases were smaller (Figs. 12 and 13). In the case of the LSTM, it also demonstrated that each reconstructed variable (salinity and temperature) requires a different tuning of the model. Overall, the RFR performed better than the LSTM with the test split dataset, but when validating with the complete reanalysis product, we observed better extrapolation and representativity of the data with the LSTM model in terms of correlation and variability assessment (Figs. 16, 18, and 19).






The contributions of each variable (SHAP values in Figs. 9 and 10) in the models have a geophysical meaning. The RFR excels at direct data relationships; thus, it uses the SSS as a base reference and modulates the variations with the rest of the variables, such as SST, latitude, or SSH. In the case of the LSTM, we can observe that it can derive those relationships with the intrinsic patterns of the data, such as the latitudinal dependence of the SST. Furthermore, it tries to balance the weight of each of the input variables, giving enough weight to each of them to make them important in the decision.

We observed some biases in the validation with the reanalysis in Fig. 15. These biases can originate from the irregular sampling of the ocean, as highly dynamic areas can attract Argo floats, making those regions more sampled. Even if the latitude and longitude coordinates are set as predictors, there is no smooth spatial transition between high-dynamic areas and calm waters. Another part of this bias is due to the well-known Bias/Variance trade-off present in these methodologies (Geman et al., 1992), where, to capture the variability of the data (which is what is interesting, as it is the dynamical part of the data), one has to deal with higher biases. However, the bias appears to have smooth spatial gradients, making it easy to study and correct in future works if needed. The absence of seasonal variation in the spatial correlation in the case of the RFR indicates that the model can represent the seasonal cycles of the variables, and the lack of variation (constant) of the time series indicates that the reconstruction is not affected by unaccounted trends. However, the LSTM model did not consider the MLD as an important variable for the reconstruction (Fig. 10), which later affects on the correlation of the reconstruction in months where the MLD is shallower (Fig. 19).

Both models were able to reconstruct the spatial and temporal variability patterns observed by the numerical model at different depths, which is a key aspect of studying the ocean's dynamics. However, we observed in Fig. 16 that both models tend to underestimate the variability range in the upper layers, losing part of the observed variability. The MSE values across different depths (Fig. 17) show consistently low values for both models throughout the different studied depths. This indicates that the model successfully captured different temporal dynamics intensities (calm waters vs. dynamical regions). This also suggests that the current sampling of the ocean provides adequate coverage of different water types for the model to learn the underlying patterns of their variability. The spatial correlations seen in Fig. 19 were also high in both models, achieving an R<sup>2</sup> score higher than 0.9 in both models in all the studied depths, indicating that the seasonal cycles and tendencies are well-captured by both of our models.

Overall, both models can capture the spatial and temporal variability of the ocean as seen by the reanalysis, with high correlations and accurate representations of seasonal cycles. However, the variability range is underestimated and should be

improved in future work. The results obtained with these models offer promising prospects for ocean reconstruction with the current observing system, and highlight the potential application of future satellite missions measuring SSS and SST simultaneously, such as the Copernicus Imaging Microwave Radiometer (CIMR) of the European Space Agency (ESA) (Donlon et al., 2023), to reconstruct, when combined with in situ profiles, a 3D reconstruction of salinity and temperature fields. However, some improvements in the specific architectures can be made. For example, we could integrate them with other architectures such as diffusion networks or encoders/decoders, which are specially used for high-resolution image generation. Using these technologies can provide a new perspective on how we observe and study the ocean.

## 445 7 Conclusions




Our study successfully demonstrated the feasibility of 4D ocean reconstruction using data-driven approaches and current observing systems, although there is still room for improvement in future work. The complexity of ocean dynamics across multiple dimensions presents significant challenges, requiring careful treatment of the input data and model architecture selection. While our models showed promising results in capturing ocean dynamics, particularly in vertical reconstruction, the horizontal dynamics reconstruction can be further improved.

Future work should focus on several aspects: investigating how to improve the temporal variability characterization, analyzing the evolution of biases to determine if constant corrections can preserve reconstructed variability, exploring alternative deep learning architectures for improved multi-dimensional reconstruction, and applying these models to real in-situ and satellite data. These findings contribute to our understanding of ocean reconstruction methodologies while highlighting the potential for further improvements in capturing the complex dynamics of ocean systems across all dimensions. This data-driven approach also contributes to further exploiting the synergy of the different and complementary ocean observation systems.

Code availability. The code used in this study is publicly available at GitHub (https://github.com/ainagarciaes/SSS-SST-4D-Reconstruction/tree/v1) under a GNU General Public License and has been archived on Zenodo (DOI: 10.5281/zenodo.11487678). The repository contains all scripts necessary to reproduce the analyses presented in this paper.

Data availability. This study has been conducted using the Copernicus Marine Service Global Ocean Ensemble Physics Reanalysis (Mercator Océan International, 2025) dataset. It is accessible through the E.U. Copernicus Marine Service Information webpage (https://data.marine.copernicus.eu/product/GLOBAL\_MULTIYEAR\_PHY\_ENS\_001\_031/description). The Argo profilers dataset (Argo, 2025) from which we derive the observed points are available through the SEANOE webpage (https://www.seanoe.org/data/00311/42182/) or all the alternative access to data options provided there. Mixed Layer Depth climatology dataset (de Boyer Montégut, 2023) can be downloaded from the Seanoe webpage https://www.seanoe.org/data/00806/91774/.

**Figure A1.** Feature importance percentage using SHAP values for RFRv1 (left) and RFRv2 (right) temperature models. Both models are trained with data that includes uncertainties.

**Figure A2.** Feature importance percentage using SHAP values for LSTMv1 (left) and LSTMv2 (right) temperature models. Both models are trained with data that includes uncertainties.

## **Appendix A: Temperature Validation**

Author contributions. Conceptualization by CGH; methology by AGE and FA. Funding and computational resources were acquired by FA. AGE did the data curation, formal analysis of the data, and visualization of the datasets and the obtained results. AGE, CGH, and FA did research and further experiments. The software was developed by AGE with the support of FA. The results were validated by AGE and CGH. This work was supervised by CGH and FA. The original draft was written by AGE and revised by CGH and FA.

Figure A3. Feature importance percentage using SHAP values for RFv2 (left) and LSTMv1 (right) temperature models without uncertainty values.

**Figure A4.** Predicted temperature vertical profiles for four different points (see Fig. 11 for its location) and the different models: RFRv1 (blue), RFRv2 (orange), LSTMv1 (green), LSTMv2 (red). The date is 29th October 2022. The black line corresponds to the OGCM baseline (the target value).

Competing interests. The authors declare that they have no conflict of interest.

Acknowledgements. This work was supported by the EO4TIP project, grant PID2023-149659OB-C21, funded by MICIU/AEI/10.13039/501100011033 and ERDF/EU. This work was also supported by the European Maritime, Fisheries and Aquaculture Fund (EMFAF). The authors would like to thank the support and computing resources from the AI4EOSC platform, which has received funding from the European Union's Horizon Europe research and innovation programme under grant agreement number 101058593. This work also acknowledges the "Severo Ochoa Centre of Excellence" accreditation, grant CEX2019-000928-S funded by MICIU/AEI/10.13039/501100011033. This work is moreover a contribution to CSIC PTI Teledetect.

**Figure A5.** Predicted temperature minus Ground Truth (50 m depth) at a  $5^{\circ} \times 5^{\circ}$  grid for (from top to bottom and left to right): RFv1, RFv2, LSTMv1, LSTMv2. The date is 29th October 2022.

**Figure A6.** Biases at different depths (from left to right: 5 m, 50 m, and 500 m) between predicted temperature and ground truth for the two proposed models RFRv2 (top row), LSTMv2 (bottom row).

**Figure A7.** Variability assessment between the predictions and the ground truth (reanalysis) temperatures at different depths (from left to right: 5 m, 50 m, and 500 m). The standard deviation of the reanalysis for the 2008-2009 period (first row). Differences between the standard deviation of the ground truth and each model, RFRv2 (second row), and LSTMv2 (third row). Solid black lines delimit statistically significant areas at a 95% confidence level assessed using an F-test for the difference of variances.

**Figure A8.** Mean Squared Error at different depths (from left to right: 5 m, 50 m, and 500 m) between ground truth and predicted temperature for the two proposed models RFRv2 (top row), LSTMv2 (bottom row).

**Figure A9.** Temporal Correlation Coefficients at different depths (from left to right: 5 m, 50 m, and 500 m) between ground truth and predicted temperature for the two proposed models RFRv2 (top row), LSTMv2 (bottom row).

**Figure A10.** Temperature spatial correlation time series using a moving average of 30 days at different depths (5m black solid line, 50 m black dashed line, and 500 m black dotted line) for the two proposed models for RFRv2 (left) and LSTMv2 (right). The average mixed layer depth is superposed in blue.

**Figure A11.** Singularity analysis of the temperature reconstruction at 500 m depth. Maps are from February 1st, 2008. Temperature (first column) and its associated singularity exponents (second column) for: the reanalysis without uncertainties (first row), RFRv2 (second row), and LSTMv2 (third row).

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
