# Peer review of "On the global reconstruction of ocean interior variables: a feasibility data-driven study with simulated surface and water column observations"

_EGUsphere, 2025_

## Author Comment (AC1)

**Reviewer 1:**

García-Espriu conduct an observing system simulation experiment (OSSE) to evaluate the feasibility of reconstructing ocean interior temperature and salinity from in situ observational data and satellite observational data products. The authors leverage output from the CMEMS Global Ocean Ensemble Reanalysis product to conduct this experiment, and they subsample the product at times and locations where Argo float profiles are available. They then use these subsampled synthetic profiles to train machine learning models, which they apply to satellite products to reconstruct ocean interior properties, and they compare these reconstructions against the reanalysis "truth" to evaluate the skill of the reconstruction methods.

The authors find that the more complex versions of their random forest regression (RFRv2) model and Long-Short Term Memory (LSTMv2) network are able to reproduce ocean temperature with an $R^2$ of around 0.85 and salinity with an $R^2$ of around 0.95. They validate their models with synthetic profiles withheld from model training and by using a regional subsection of the reanalysis dataset. They report validation statistics spatially and by depth, concluding that the RFRv2 model performed better in terms of the evaluation statistics against the test dataset but the LSTMv2 model was better able to represent the data in terms of variability over time and space. The authors also use SHapley Additive exPlanations (SHAP) to interpret their trained models.

Overall, I support the approach this manuscript takes to the question of how in situ and satellite observing systems can be leveraged to reconstruct ocean interior properties. However, it falls short in its execution and interpretation of the analysis. Most importantly, the authors could attempt to remedy or discuss more extensively the shortcomings of the models to predict ocean interior variables from primarily surface data, and the results could be better placed into context among similar studies that reconstruct ocean interior properties from observational data.

We would like to thank the reviewer for her/his valuable comments. Our response is given in blue, and the number of lines corresponds to those of the new manuscript with track changes.

General suggestions

One aspect that I think is missing from the manuscript is the contextualization of the authors' results with similar methodologies that have been applied to map salinity and temperature from observations (a few of which are referenced in the introduction). Although not all studies that reconstruct ocean interior properties from observations include a reanalysis-based evaluation of mapping accuracy (as is the focus of this manuscript), many report error statistics of their reconstructions evaluated against independent data. Su et al. (2018), for example, evaluate their reconstructed subsurface temperature anomalies using root mean squared error and $R^2$ as metrics, and the results of the OSSE reported here could be evaluated against those results.

We have included some contextualization in the discussion section as suggested by the reviewer. (lines 403-412)

In general, I was surprised to see such high disagreement with the test data at depth, when temperature and salinity should be more constant in space and time, and therefore relatively easier to reconstruct than at the surface. Buongiorno Nardelli (2020), for example, retrieves minimum errors for temperature and salinity at depth. This, in my opinion, points to an aspect of the methodology that can be significantly improved. It is not particularly surprising that a model based primarily on surface characteristics would struggle to estimate temperature and salinity at 1000 meters. I suspect a strategy of somehow de-emphasizing the impact of the surface predictor datasets as depth increases

might improve these high offsets at depth. In any event, this is another instance where contextualization of the results of this OSSE would be helpful.

We understand the concern of the reviewer. The main question of this work, as he /she pointed out previously, is to assess how in situ and satellite observing systems can be leveraged to reconstruct ocean interior properties. However, the in situ profiles are only used for training our models, and then the reconstruction of subsurface fields is done using only surface fields. This is why our metrics degrade with depth. We have tried to clarify this point in the new version of Figure 3 and make the text clearer (lines 170-174). In addition, we contextualized the results in the discussion section (lines 403-411) and pointed out other possible strategies to improve this point in further work.

Lastly, the authors miss an opportunity to incorporate uncertainties into their experiment, or at least to discuss their implications. OSSEs present an opportunity to mimic real-world conditions; in reality, satellite observations are not perfect, nor are temperature and salinity measurements from profiling floats. Incorporating measurement uncertainty estimates in the analysis would be an important piece for answering the central question of how feasible it is to use satellite and in situ data to reconstruct ocean interior properties.

We would like to thank the reviewer, especially for this point. We have re-done the study, taking into account the typical uncertainties of each observable variable used in the study (taken from some of the products detailed in tables 1 to 3), and repeated the validation of the models. We provide a comparison of the general metrics with and without uncertainties in Table 5. We could conclude that even if the metrics are slightly degraded when uncertainties are included, as expected, the main conclusions of our work are still valid. We have modified the text accordingly.

Line-by-line comments

**Abstract: I would suggest defining the simulated in situ measurement platforms as "Argo floats" or "profiling floats" rather than buoys in the abstract.**

Following the reviewer's comment, we have changed both appearances of "*buoys*" to "*Argo floats*".

**28: Presumably, this should say "subsurface temperature and salinity"?**

The reviewer is right. We changed the text from *"subsurface temperature and subsurface temperature anomalies"* to *"subsurface temperature and salinity anomalies"*.

**86: Awkward phrasing in reference to the equatorial region.**

We changed the wording to "*equatorial region*".

**97: punctuation issue here.**

We updated accordingly.

**161-166: I'm not sure I understand the training and test split. Are you withholding some percentage of the dataset on a daily frequency (if so, what percentage?) for testing during model training? How does this differ from the ground truth dataset that is being used for evaluation?**

Our input data are only the vertical profiles of the reanalysis model (with their associated surface information) for the points where there was an Argo profile that day for the 2010-2022 period. This data is stored as daily files, which are then divided into an 80/20 split. The rest of the points of the reanalysis, where there were no Argo profiles registered, can be used for validating the model, as

they will not be seen in the training of the models. However, we further independentized it in the validation section and used the 2008-2009 period, but we could have made it using any period of the 2010-2022 as most of the points are not seen by Argo and thus, not included in the training of the models.

We have changed Fig. 3 to clarify how the datasets are constructed and how the train/test split was divided. The text explaining the separation into train and test splits has also been updated and now contains the following:

*"Finally, we separate our datasets into a train/test split, which will be common for all the trained models. This separation is made using an 80/20 ratio, where 80\% of data will be used for training and 20\% for validation as usual in machine learning models. We generate one dataset (or datafile) for each day. As the objective of our study is to analyze the feasibility of the reconstruction using current sampling of the ocean (and not predicting future trends and events), the separation is done by randomly separating the dates, but ensuring that each month is represented equally in both datasets. This avoids adding imbalances due to seasonal cycles that must be accounted for."*

[Figure]

**Figure 3.** Observing system datasets generation from  Copernicus Marine Service reanalysis data using Argo floats and surface satellite measurements sampling. For each day, the 10-day windowed simulated profiles are collocated with the central date data, generating a daily array of synthetic profiles. The different colors indicate different days in the 10-day window of a specific central date.

**173: It would be helpful to specify the metric you are referring to when discussing "accuracies".**

We changed "*Table 1 shows each model's accuracy and error metrics*" to "*Table 1 shows each model's accuracy ($R^2$) and error metrics (MSE, MAE)*" to make it clearer when reading.

**239: What is meant by "it does not overlap with the training dataset"? There are no Argo profiles from 2008-2009 in this region?**

We used the 2010 to 2022 Argo floats for the training of our models because, from 2010 onwards, the number of profiles increased significantly. The specific filtering criterion and time overage for the training dataset are specified in Section 2.

*"... We use all available profiles from 2010 to 2022, but only consider those that reach a minimum depth of 1000 meters and have good quality measurements according to their quality control standard."*

**272: should be "…each of them with their own…"**

We updated the phrase accordingly.

---

## Author Comment (AC2)

**Reviewer 2:**

Review of the paper "*On the reconstruction of ocean interior variables: a feasibility data-driven study with simulated surface and water column observations*"

by Aina García-Espriu, Cristina González-Haro, and Fernando Aguilar-Gómez

This study investigates the feasibility of reconstructing ocean interior variables, specifically temperature and salinity profiles, using AI-based algorithms applied to simulated satellite surface data and in situ buoy observations. Leveraging an Observing System Simulation Experiment (OSSE) with outputs from a numerical ocean reanalysis model from the EU Copernicus Marine Service, the authors compare the performance of Random Forest Regressors (RFR) and Long-Short Term Memory (LSTM) networks. The results show that both models reasonably capture the spatial and temporal variability of ocean interior conditions (particularly for salinity), with RFR offering higher accuracy in direct reconstructions and LSTM demonstrating better extrapolation capabilities with ground truth observations. The findings highlight the potential of data-driven approaches to enhance 4D ocean reconstruction and contribute to future digital twin ocean frameworks, while also identifying current challenges in capturing vertical variability and reducing biases. Nevertheless, the study lacks some aspects that should be integrated at least at the discussion level.

We would like to thank the reviewer for her/his valuable comments. Our response is given in blue, and the number of lines corresponds to those of the new manuscript with track changes.

Major Points

**I could not fully understand how surface information is synthesized from the Copernicus Marine Service numerically modelled data. To the best of my understanding, the aim is to provide insights on a potential 4D reconstruction that exploits satellite-based surface observations. In particular, the Authors claim the intention to perform reconstructions at the spatial resolution provided by space-based microwave sensors. However, it seems surface observations are directly extracted from modelled surface data. To be consistent, an assessment of the type and effective resolutions of satellite input data should be performed, and the synthetic input data should be adjusted accordingly. For example, present-day satellite-based sea surface heights/currents/temperature could differ significantly with respect to the outputs of a hydrodynamic model. A discussion on how this could impact the results of the 4D reconstruction could be beneficial.**

We thank the reviewer for this comment; it made us realize that the text of the manuscript was not clear enough. The main question of this work is to assess how in situ and satellite observing systems can be leveraged to reconstruct ocean interior properties. To do so, we based our study on an Observing System Simulation Experiment (OSSE) in which we use the outputs from an ocean numerical model as the ground truth, and simulate a real observing system of the ocean, taking the surface of the model as a simulation of satellite observations, and vertical profiles in the same locations as the real Argo floats. We have tried to clarify this point in the new version of Figure 3 and make the text clearer (lines 181-197). In addition, we have included a new section, 2.1 Surface Remote sensing products, that summarizes the main spatial and temporal resolutions and uncertainties of some of the available remote sensing products for the variables we use in our study.

**On the same note, I think the paper lacks discussions on the capability of current satellite missions and, more importantly, future missions for Earth observations in the microwave band, how this could impact e.g. sea surface temperature and salinity monitoring and which could be the impact of such missions on the proposed ocean 4D reconstruction. I think this should also be integrated into the discussion section, at least.**

We agree with the reviewer; we have included a small paragraph in the discussion section to contextualize our approach and how it could benefit from new simultaneous SST and SSS observations provided by the CMIR mission. (see lines 446-449)

Minor Points

**I was wondering if the proposed reconstruction methodology is able to provide un uncertainty estimate to verify if the profiles provided in Figure 10 can be considered significantly different. Could the Authors quickly comment on that?**

Currently, our model does not provide uncertainty estimates for the reconstruction. Our primary objective was to assess the feasibility of the reconstruction approach rather than to deliver a ready-to-use data product. At this stage of development, we did not consider uncertainty quantification to be essential for demonstrating the methodology's viability. That said, we fully acknowledge that uncertainty estimation will be crucial for future work aimed at producing an operational data product and should be incorporated in practical applications.

**Have the Authors tried to inter-compare the feature-resolution of the reconstructed fields versus the ground truth? Are you expecting significant differences?**

We have included a new section (5.4 Spatio-Structural Validation) in which we further qualitatively discuss the feature-resolution using the singularity analysis, and we show that the LSTM gives a better feature reconstruction when comparing with the ground-truth (lines: 380-389)

**Could the Authors also provide a broad overview of which could be the "real in-situ and satellite" data more suitable for their future applications?**

We have included a new section (2.Current Sampling of the Ocean) where we summarize the current satellite and real in situ products available, regarding the spatial and temporal resolution, and the uncertainties of each variable we used in our study.

**Typos**

**In general, please always use Copernicus Marine Service instead of CMEMS when referring to data generated within the EU Copernicus Marine Service.**

We updated all the appearances in the text.

**Line 88: earth-> Earth**

We updated the word accordingly.

---

## Referee Report (RR1)

Review of revised version of **On the reconstruction of ocean interior variables: a feasibility data-driven study with simulated surface and water column observations** A. García-Espriu, C. González-Haro, F. Aguilar-Gómez submitted to *EGUsphere*

**General comments**

I thank the authors for their responses to my initial round of comments. The revised manuscript is much improved and explores the results obtained from the OSSE in greater detail. The addition of Figure 3 is very helpful in describing the approach employed in the study. The added material in the discussion section to contextualize the results relative to similar studies is valuable as well.

I appreciate the authors taking the opportunity to simulate uncertainties in their reconstruction experiments. I am, however, unclear about the statement in lines 183–185 that simulating uncertainties as Gaussian noise is a "worst-case-scenario" for uncertainty estimates. I'd think that consistent Gaussian noise is actually a best-case scenario. A worst-case would consider possible systematic biases and/or noise that is not consistent in space or time. These represent additional uncertainty scenarios that could be explored, but are probably outside the scope of this study.

Overall, I support the acceptance of this manuscript subject to minor revisions.

**Line-by-line comments**

Figure 8: Typo – should be LSTMv2

Figures 9 and 10: It would be helpful to specify in these figure captions that these models are trained with data that include uncertainties.

Figure 12: Define the black OGCM line in the figure caption as well.

316–317: It would be helpful to point to a figure in referencing the sampling density and dynamic environment over which this validation is performed.

376: Note in making this point that the LSTMv2 gives a better feature reconstruction "than the RFRv2"

Figure 20. This figure is not referenced in the text. The discussion in Section 5.4 should be expanding with more information of exactly what the singularity exponent analysis is telling us about the reconstructions.

---

## Author Response (AR2)

I thank the authors for their responses to my initial round of comments. The revised manuscript is much improved and explores the results obtained from the OSSE in greater detail. The addition of Figure 3 is very helpful in describing the approach employed in the study. The added material in the discussion section to contextualize the results relative to similar studies is valuable as well.

We would like to thank the reviewer once again for her/his valuable comments. Our response is given in blue, and the number of lines corresponds to those of the new manuscript with track changes.

I appreciate the authors taking the opportunity to simulate uncertainties in their reconstruction experiments. I am, however, unclear about the statement in lines 183–185 that simulating uncertainties as Gaussian noise is a "worst-case-scenario" for uncertainty estimates. I'd think that consistent Gaussian noise is actually the best-case scenario. A worst-case would consider possible systematic biases and/or noise that is not consistent in space or time. These represent additional uncertainty scenarios that could be explored, but are probably outside the scope of this study.

We do agree with the reviewer. We have rephrased lines 185-190 accordingly to make that point clearer.

Overall, I support the acceptance of this manuscript subject to minor revisions.

Line-by-line comments

Figure 8: Typo – should be LSTMv2

Typo has been addressed.

Figures 9 and 10: It would be helpful to specify in these figure captions that these models are trained with data that include uncertainties.

We specified the usage of uncertainties in the captions of Figures 9, 10, and A1, A2.

Figure 12: Define the black OGCM line in the figure caption as well.

The caption has been modified to define the black line as suggested.

316–317: It would be helpful to point to a figure in referencing the sampling density and dynamic environment over which this validation is performed.

We thank the reviewer for this comment. We have pointed to Fig. 2, where we show the sampling of Argo floats and the temporal standard deviation of SST and SSS.

376: Note in making this point that the LSTMv2 gives a better feature reconstruction "than the RFRv2" Added.

Figure 20. This figure is not referenced in the text. The discussion in Section 5.4 should be expanding with more information of exactly what the singularity exponent analysis is telling us about the reconstructions.

We fixed the reference to Figure 20 (it referenced another figure that used the same label). We have added some lines of discussion on the exponents in lines 380-381.